# Effect of Probiotic Lactic Acid Bacteria (LAB) on the Quality and Safety of Greek Yogurt

**DOI:** 10.3390/foods11233799

**Published:** 2022-11-25

**Authors:** So-Young Yang, Ki-Sun Yoon

**Affiliations:** Department of Food and Nutrition, College of Human Ecology, Kyung Hee University, 26 Kyungheedae-ro, Dongdaemun-gu, Seoul 02447, Republic of Korea

**Keywords:** Greek yogurt, *Lactobacillus gasseri* BNR17, *Lactobacillus plantarum* HY7714, enterohaemorrhagic *E. coli*, preference test

## Abstract

Greek yogurt is a strained yogurt with a high protein content that brings nutritional benefits. To enhance the functional benefits of Greek yogurt, Greek yogurt was prepared with various combinations of probiotic lactic acid bacteria (LAB) (*Streptococcus thermophilus*, *Lactobacillus bulgaricus*, *Lactobacillus gasseri* BNR17, and *Lactobacillus plantarum* HY7714). Effects of probiotic LAB on quality, sensory, and microbiological characteristics of Greek yogurt were then compared. Among samples, Greek yogurt fermented by *S. thermophilus* and *L. bulgaricus* showed the highest changes of pH and titratable acidity during 21 d of storage at 4 °C. Greek yogurt fermented with *L. plantarum* HY7714 had a higher viscosity than other samples. Greek yogurt fermented with *S. thermophilus*, *L. bulgaricus*, *L. gasseri* BNR17, and *L. plantarum* HY7714 showed superior physicochemical properties and received the highest preference score from sensory evaluation among samples. Overall, the population of enterohaemorrhagic *Escherichia coli* (EHEC) was more effectively reduced in Greek yogurt fermented with probiotic LAB than in commercial Greek yogurt during storage at 4, 10, and 25 °C. Thus, the addition of *L. gasseri* BNR17 and *L. plantarum* HY7714 as starter cultures could enhance the microbial safety of Greek yogurt and sensory acceptance by consumers.

## 1. Introduction

Greek yogurt is known as a healthy snack that can increase lean muscle mass and decrease body fat [1]. The manufacture of Greek yogurt begins with homogenization of standardized milk. The homogenized milk is then pasteurized and cooled at an incubation temperature up to 40 °C. After a starter culture is inoculated, the yogurt gets a thicker texture through a concentration step [1,2]. This “concentration” step can increase protein content to be around 9–10% and give Greek yogurt a hard texture [3]. Protein in Greek yogurt makes the perception of hunger lower and the time between meals longer [4].

Commercial yogurt is prepared with probiotics to help intestinal function, stabilize gut microflora, and change compositions and numbers of intestinal microflora [5]. *Lactobacillus gasseri* BNR17 originally isolated from human breast milk is known to inhibit weight gain [6]. *L. gasseri* BNR17 can reduce the amount of food intake and 2 h postprandial blood glucose [7]. *L. gasseri* BNR17 can also reduce levels of leptin and insulin [8], waist circumferences, hip circumferences, and visceral adipose tissues [9]. *Lactobacillus plantarum* HY7714 isolated from healthy infant feces is a registered probiotic that can prevent photoaging, restore procollagen, and increase the retention of water in the face and hand [10,11,12,13]. Recently, Lee et al. [14] have reported that *L. plantarum* HY7714 can produce exopolysaccharides, which can control tight junctions in intestinal epithelial cells and recover cytotoxicity and hydration capacity in Hs68 cells induced by UVB irradiation. Although *L. gasseri* BNR17 and *L. plantarum* HY7714 are used to manufacture functional capsules or powder, neither *L. gasseri* BNR17 nor *L. plantarum* HY7714 has been applied in the manufacturing of healthy foods such as yogurt.

Enterohaemorrhagic *Escherichia coli* (EHEC) has strong acid resistance and maintains viability at low pH [15]. Acid resistance of EHEC has been clarified more due to the outbreak of EHEC in yogurt, where the risk of foodborne pathogens is very low [16]. EHEC can survive well in yogurt at 4 and 10 °C for 21 d of storage [17]. Recently, the antimicrobial effect of lactic acid bacteria (LAB) on *E. coli* has been reported [18,19]. LAB shows antagonistic activities, expressing higher inhibition effects on *E. coli* and *E. coli* O157:H7 than *Salmonella* typhimurium, *S. Enteritidis*, and *Listeria monocytogenes* [18]. Choi et al. [19] have reported that *Leuconostoc mesenteroides* and *L. plantarum* can inhibit the viability of *E. coli* O157:H7 in kimchi. Although consumers’ preference for Greek yogurt is increasing, *L. gasseri* BNR17 and *L. plantarum* HY7714 have not been tested as starter cultures for Greek yogurt manufacturing. Their effects as antimicrobial agents in various functional foods remain unclear.

Therefore, the objective of this study is to manufacture new functional Greek yogurt with *L. gasseri* BNR17 and *L. plantarum* HY7714 known to have various health benefits. How each LAB affected the viscosity, consumer preference, and microbiological safety of Greek yogurt at 4, 10, and 25 °C is also investigated.

## 2. Materials and Methods

### 2.1. Starter Culture for Preparation of Greek Yogurt

For Greek yogurt production, probiotic LAB including *Streptococcus thermophilus* (ST, KCTC 3779), *Lactobacillus bulgaricus* (LB, KCTC 3635), *Lactobacillus gasseri* BNR17(KCTC 10902BP), and *Lactobacillus plantarum* HY7714 (KCTC 12400BP) were purchased from Korean Collection for Type Cultures (KCTC). All strains were maintained at −80 °C in De Man, Rogosa, and Sharpe broth (MRS, Difco^TM^, Difco Laboratories, Detroit, MI, USA) with 20% glycerol. Thawed ST was aerobically incubated at 42 °C overnight in an incubator. LB was anaerobically incubated overnight at 36 °C using an anaerobic jar system (Don Whitley Scientific Ltd., Bradford, UK). *L. gasseri* BNR17 and *L. plantarum* HY7714 were aerobically incubated at 36 °C with shaking at 140 rpm in a rotary shaker (VS-8480SR, Vision). *L. gasseri* BNR17 and *L. plantarum* HY7714 were centrifugated at 4500× *g* for 15 min (VS-550, Vision), washed with sterile phosphate-buffered saline (PBS) twice, and resuspended to a final concentration of 10^9^ CFU/mL according to Kim et al. [11] with some modifications.

### 2.2. Preparation of Enterohaemorrhagic E. coli for the Safety Study

Enterohaemorrhagic *E. coli* (EHEC) strains (NCCP 13720, 13721), including *E. coli* O157:H7 (NCTC 12079), were obtained from the Ministry of Food and Drug Safety (MFDS) to investigate the effect of LAB probiotics on the behavior of EHEC in Greek yogurt at various temperatures. Frozen strains were maintained at −80 °C with 20% glycerol in tryptic soy broth (TSB, MB cell, Seoul, Republic of Korea). Then 10 μL of thawed EHEC was inoculated into 10 mL of TSB and incubated at 36 °C in a rotary shaker at 140 rpm overnight. After centrifuging at 4000× *g* for 10 min, the supernatant was removed, and the pellet was washed with 10 mL of 0.1% sterilized peptone water (Difco^TM^ Peptone water, Difco Laboratories). A cocktail of EHEC strains was prepared by resuspending them with 0.1% sterilized peptone water. Then 1 mL aliquot of EHEC was diluted with 0.1% sterilized peptone water for inoculum.

### 2.3. Manufacturing of Greek Yogurt

Pasteurized whole milk by high-temperature short-time (HTST) was purchased from a local market (Seoul, Republic of Korea) and heated in glass bottles at 42 °C in a water bath (SB-1200, EYELA Co., Ltd., Shanghai, China). Four different Greek yogurts with various combinations of starter culture (5% wt/wt) were prepared as follows: ST and LB as traditional yogurt strains (T1, control), ST, LB, and *L. gasseri* BNR17 (T2), ST, LB, and *L. plantarum* HY7714 (T3), and ST, LB, *L. gasseri* BNR17, and *L. plantarum* HY7714 (T4). All strains were mixed in equal proportions. Fermentation of Greek yogurt was carried out at 42 °C until pH was 4.4. Samples were then distributed into 250 mL polypropylene centrifuge bottles (Beckman Coulter Avanti^TM^, Indianapolis, IN, USA) and centrifuged at 4500× *g* for 15 min. After Greek yogurt (60 g) was transferred to sterilized plastic cups, quality evaluation was conducted every three days for 21 days of storage at 4 °C. A flow chart of the manufacturing step for Greek yogurt is shown in Figure 1.

### 2.4. pH, Titratable Acidity, and Viscosity

The pH and titratable acidity were measured according to the method of AOAC [20]. Greek yogurt (10 g) was homogenized with 90 mL of distilled water using a stomacher (Stomacher, Interscience, Saint-Nom-la-Bretèche, France). The pH was measured with a benchtop pH meter equipped with a glass electrode (OrionTM Star A211, Thermo Fisher Scientific Co., Waltham, MA, USA). Greek yogurt (5 g) was homogenized with 45 mL of distilled water in a sterile filter bag to determine titratable acidity. Then 20 g of the filtrate was titrated with 0.1 N NaOH using 0.5 mL of phenolphthalein indicator until pH reached 8.3. The titratable acidity was expressed as a percentage of lactic acid. It was calculated as follows:(1)$$\mathrm{Titratable}\;\mathrm{acidity}\;\left(\%\right)=\frac{0.1\;N\;\mathrm{NaOH}\;\left(\mathrm{mL}\right)\times 0.1N\;\mathrm{NaOH}\;\mathrm{factor}\times 0.009}{\mathrm{Sample}\;\left(g\right)}\times 100.$$

The viscosity of Greek yogurt was determined according to the method of Ghasempour et al. [21] with modifications. Briefly, all samples were divided into glass bottles in equal volumes. The viscosity of each sample was measured with RV-spindle No. 6 for 20 s at 10 rpm using a Brookfield viscometer (DV1, Brookfield Laboratories, Inc., Middleboro, MA, USA). It was expressed as Pascal-second (Pa·s) and millipascal-second (mPa·s).

### 2.5. Consumer Prefernce Test

The consumer preference test was performed following the rules of the Institutional Review Board (IRB) to comply with bioethics (KHSIRB-21-354). Four Greek yogurt samples (T1: ST and LB; T2: ST, LB, and *L. gasseri* BNR17; T3: ST, LB, and *L. plantarum* HY7714; T4: ST, LB, *L. gasseri* BNR17, and *L. plantarum* HY7714) were stored at 4 °C before the sensory test. Each sample (30 g) was scooped into paper cups labeled with 3-digit random numbers, which were served monadically to the panelist with spring water and a spoon. The consumer preference test was conducted by 60 panelists (44 women and 16 men) aged between 20 and 68. The purpose of sensory evaluation and the direction of how to score the sensory properties of samples (flavor, sweetness, sourness, viscosity, creaminess, mouthfeel, and overall acceptance) were provided to participants (Table 1). Participants scored each sample with a 7-point Hedonic scale (1 = dislike very much; 2 = dislike moderately; 3 = dislike slightly; 4 = neither like nor dislike; 5 = like slightly; 6 = like moderately, 7 = like very much) [22].

### 2.6. Microbiological Analysis

Enumeration of LAB in Greek yogurt was carried out by the standard plate counting method. Each sample (10 g) was diluted 10-fold with 0.9% sterile saline solution (NaCl, Duksan, Ansan-si, Republic of Korea). Then 1 mL of aliquot was inoculated onto MRS agar and incubated at 36 °C for 48 h.

To analyze the behavior of EHEC in Greek yogurt, each sample made with different probiotic LAB was aseptically divided in 10 g into 50 mL conical tubes (SPL Life Science Co., Pocheon-si, Republic of Korea) and compared with a commercial Greek yogurt (Foodis Plain Greek yogurt, ILDONG Foodis Co., Seoul, Republic of Korea) as a control. The commercial Greek yogurt was manufactured with complex lactic acid bacteria (15 × 10^10^/80 g). Each Greek yogurt was inoculated with a cocktail of diluted EHEC at an initial level of 5~6 log CFU/g and stored at 4, 10, and 25 °C. After an appropriate interval time, samples were homogenized with 0.1% sterilized peptone water, and 1 mL of aliquot was serially diluted. A diluted solution of EHEC was spread on eosin methylene blue agar (EMB agar, Oxoid) and incubated at 36 °C for 24 h to analyze the behavior of EHEC in Greek yogurt. The primary survival model of EHEC in Greek yogurt was applied to the Weibull model [26] (Equation (1)) using the GinaFit V1.7 program [27]. Delta value (time for the first decimal reduction) was then calculated.
(2)$$\mathrm{Weibull}\;\mathrm{equation}:\;\mathrm{Log}\left(N\right)=\mathrm{Log}\left(N_{0}\right)-{\left(\frac{t}{delta}\right)}^{p}$$

*N*_0_: log the initial number of cells.

*t*: time.

*delta*: time for the first decimal reduction.

*p*: shape (*p* > 1: concave downward curve; *p* < 1: concave upward curve; *p* = 1: log-linear).

The secondary model for delta value was developed, and the Davey model (Equation (3)) was used to predict delta values as a function of temperature:(3)$$\mathrm{Davey}\;\mathrm{model}:\;Y=a+\left(b/T\right)+(c/T^{2})$$

a, b, c: constant.

T: temperature.

### 2.7. Statistical Analysis

All experiments were conducted three times or more. Results of this study were subjected to ANOVA and Duncan’s multiple range test using SAS software ver. 9.4 (SAS Institute, Inc., Cary, NC, USA). The significance was tested at *p* < 0.05 level.

## 3. Results and Discussion

### 3.1. Effect of Probiotic LAB on pH and Titratable Acidity

Changes in physicochemical characteristics of Greek yogurt at 4 °C during 21 d of storage are shown in Figure 2 and Figure 3. The pH and titratable acidity of traditional Greek yogurt were 4.06 to 4.64 and 0.919 to 1.579%, respectively [28]. High acidity can negatively affect the water-holding capacity and viscosity of yogurt [29]. After centrifugation, the pH values of all samples decreased from 4.4~4.44 to 4.15~4.17 in this work. The pH of the Greek yogurt made with ST and LB (T1) was further dropped from 4.167 to 3.843 after 21 d of storage, while Greek yogurt made with ST, LB, and *L. gasseri* BNR17 (T2) had relatively constant pH values (Figure 2). Overall, the pH decrease rate was low in Greek yogurt containing *L. gasseri* BNR17 (T2 and T4) during 21 d of storage at 4 °C (Figure 3). The titratable acidity of all Greek yogurt samples increased during 21 d of storage. The increase of titratable acidity of Greek yogurt made with ST and LB (T1) was the highest (+0.780 ± 0.227) among all samples (*p* < 0.05) during storage. The pH dropped while titratable acidity increased in all Greek yogurt samples during storage in this study, similar to the results of previous studies [30,31]. Greek yogurt made with *L. gasseri* BNR17 showed the least changes in pH and titratable acidity during storage, indicating that *L. gasseri* BNR17 did not affect the pH change of yogurt. Increased levels of *L. rhamnosus* GG decrease titratable acidity [32]. *L. casei* AST18 also inhibits the acidogenicity of a commercial yogurt starter [33]. These results confirm that changes in the physicochemical characteristics of Greek yogurt can be controlled by the kind of starter culture used during yogurt manufacturing. Adding various probiotic LAB as starter cultures may contribute to the development of sensory characteristics of a consumer-oriented product.

### 3.2. Effect of Probiotic LAB on Viscosity and LAB Population

Consumers favor Greek yogurt because of its unique firmness, dense texture, moderate sweet aromatic, milk fat and dairy sour flavors, and moderately sour taste [23]. Thus, viscosity is one of the most important quality characteristics of Greek yogurt. In the present study, the viscosity of Greek yogurt ranged from 70.9 to 71.4 Pa·s (7.09 × 10^4^ to 7.14 × 10^4^ mPa·s) at 0 d after centrifugation. It was increased during 21 d of storage at 4 °C (Figure 2). Greek yogurt containing *L. plantarum* HY 7714 (T3 and T4) had the highest viscosity at 21 d and the highest increase rate of viscosity during 21 d of storage among samples (Figure 3) (*p* < 0.05). *L. plantarum* strains can produce exopolysaccharide (EPS), a macromolecule composed of monosaccharide residues of sugar and sugar derivatives. EPS can act as an important factor in the physicochemical and rheological properties of yogurt due to its role as a natural concentrate agent and stabilizer [34,35]. Nambiar et al. [36] have reported that EPS isolated from *L. plantarum* HM47 has high thermal stability and that it can enhance the texture of yogurt at a low pH (4.0). In this work, *L. plantarum* HY 7714 also increased the viscosity of Greek yogurt.

As shown in Figure 2, Greek yogurt had LAB populations above 10^8^ CFU/g after a concentration step. Yogurt should contain a minimum of 10^7^ CFU/g of live and active cultures [37]. The LAB population was well maintained above the criteria in Greek yogurt at 4 °C during 21 d of storage in this work. Greek yogurt with ST, LB, and *L. gasseri* BNR17 (T2) showed the lowest decrease in the LAB population (−0.022 ± 0.002) during 21 d of storage at 4 °C. However, no significant difference in the decreased extent of LAB population was observed between T2 and T4 (ST, LB, *L. gasseri* BNR17, and *L. plantarum* HY7714) (Figure 3). These results indicated that populations of probiotic LAB were well maintained in Greek yogurt containing *L. gasseri* BNR17 during storage.

### 3.3. Antimicrobial Effect of Probiotic LAB on EHEC

Effects of LAB on the survival of EHEC in Greek yogurt stored at 4, 10, and 25 °C are shown in Figure 4. At both 4 °C and 10 °C, the most rapid reduction of EHEC was observed in Greek yogurt made with ST and LB (T1). On the other hand, populations of EHEC were well maintained in all other Greek yogurt stored at 4 °C and 10 °C. At 25 °C, the most rapid reduction of EHEC was observed in Greek yogurt made with ST, LB, and *L. plantarum* HY7714 (T3), in which EHEC was not detected after 4 d of storage. The population of EHEC was maintained in commercial Greek yogurt up to 4 d of storage and then rapidly decreased, indicating that the type of probiotic LAB could affect the behavior of EHEC in Greek yogurt at ambient temperature. Ogwaro et al. [38] have manufactured yogurt with pasteurized full cream milk and found that *E. coli* O157:H7 can survive at 4 °C, while there is no *E. coli* O157:H7 on the fifth day of storage time at 25 °C. Moineau-Jean et al. [39] have also prepared Greek-style yogurt using centrifugation and ultrafiltration methods. Ultrafiltration methods more effectively inhibited the viability of non-pathogenic *E. coli* strains than traditional and centrifugation methods. In addition, *E. coli* had a lower viability at 8 °C than at 4 °C (*p* < 0.05).

Commercial Greek yogurt had higher delta values than other Greek yogurt prepared in this work (*p* < 0.05), indicating that the highest survival ability of EHEC was observed in commercial Greek yogurt at all temperatures. This trend was confirmed with delta values as a function of temperature, which indicated how rapidly pathogens were killed with an increase in temperature (Table 2). The low pH (4.52) and viscosity (59.6 ± 0.26 Pa·s, 5.96 × 10^4^ ± 0.26 mPa·s) of commercial Greek yogurt might have affected the survival ability of EHEC. On the contrary, EHEC died more quickly in Greek yogurt prepared with various probiotic LAB in the present study. Especially, Greek yogurt made with ST and LB (T1) had the lowest pH and delta values and the highest titratable acidity among all samples (*p* < 0.05). The presence of lactic acid produced from LAB can control the growth of *E. coli* O157:H7 in yogurt [40]. Hu et al. [41] have observed that organic acid produced from *L. plantarum* exhibits antimicrobial activity against *E. coli*. *E. coli* O157:H7 was also inhibited by reduced pH and lactic acid produced by *L. acidophilus* and *L. casei* [42]. Guraya et al. [43] have reported that a pH below 4.1 can significantly inhibit the growth of EHEC in yogurt. Moreover, cell-free supernatant of LAB strains, including *Leuconostoc mesenteroides* and *L. plantarum* shows antimicrobial activities against EPEC, ETEC, and *E. coli* O157:H7 in kimchi [19].

### 3.4. Consumer Preference Test

Consumer preference test results of Greek yogurt are shown in Table 3. There were no significant differences in flavor scores of Greek yogurts among all samples, indicating that the panelist did not recognize the difference in the flavor of Greek yogurts prepared in this work. The average preference score for the flavor of Greek yogurts ranged from 4.82 to 5.08. Overall, Greek yogurt made with ST, LB, *L. gasseri* BNR17, and *L. plantarum* HY7714 (T4) had the highest preference scores of all sensory properties, including flavor (5.08), sweetness (5.05), sourness (5.3), viscosity (5.42), creaminess (5.68), mouthfeel (5.87), and overall acceptability (5.8). Among sensory preference scores, scores for the sourness and overall acceptability of Greek yogurt (T4) were significantly higher than those of other Greek yogurt samples (*p* < 0.05). For sourness, T4 (5.3) had the highest preference score, followed by T2 (4.75), T3 (4.42), and T1 (4.37) (*p* < 0.05). These results are related to the titratable acidity of Greek yogurt. The highest titratable acidity of Greek yogurt made with ST and LB (T1) received the lowest preference score for sourness. Greek yogurt containing *L. gasseri* BNR17 or/and *L. plantarum* HY7714 (T2, T3, T4) received higher scores of sweetness and sourness than Greek yogurt containing only traditional starter culture (T1). Although there were no significant differences in viscosity scores among samples, T4 had the highest viscosity score (5.42) among all samples, followed by T3 (5.28), T2 (5.25), and T1 (5.15). T4 also had the highest scores for creaminess (5.68) and mouthfeel (5.87), followed by T3 (5.55 and 5.57), T2 (5.4 and 5.4), and T1 (5.18 and 5.3). Lastly, Greek yogurt containing both *L. gasseri* BNR17 and *L. plantarum* HY7714 (T4) had the highest score for overall acceptability (5.8), which indicates “like moderately”, followed by T2 and T3 with the same score (5.2) and T1 (4.78) (*p* < 0.05). These results show that using *L. gasseri* BNR17 and *L. plantarum* HY7714 as starter cultures in Greek yogurt manufacturing can improve various sensory qualities of yogurt, which are closely related to consumer preference and acceptability. Moreover, the highest preference sensory score for T4 among samples might be attributed to the combination of various probiotic LAB in T4. Coggins et al. [44] have found that taste and texture rather than flavor or appearance make a difference in the preference for yogurt. Aroma, sweetness, sourness, chalky mouthfeel, and viscosity are also significant factors affecting the preference for yogurt drinks [45].

LAB can change carbohydrates into lactic acid or other metabolites, caseins into peptides and free amino acids, and milk fat into free fatty acids during fermentation. These mechanisms make the unique flavor of yogurt [46,47]. *Leuconostoc* strains are preferred to increase the butter-like flavor of yogurt due to diacetyl, acetic acid, and ethanol produced during fermentation [48]. Bifidobacteria contribute to the production of acetaldehyde and acetoin with effects on the overall flavor quality of yogurt [49]. When quantitative descriptive analysis and consumer preference evaluation were performed to compare six conventional yogurt samples and three probiotic yogurt samples, probiotic yogurt samples had higher scores of sweet taste, creaminess, and overall sensory quality than conventional yogurt samples. This result appeared to be due to the high preference for probiotic yogurt samples for the degree of uniformity of particles and viscosity in the mouth [50]. Recently, higher preferences for color and overall taste of probiotic yogurt samples containing *L. fermentum* KU200060 than control yogurt have been also reported [51]. Additionally, a combination of *L. rhamnosus* GG, *L. plantarum* NK181, or *L. delbeuckii* KU200171 with a traditional starter culture led to high scores of tastes, texture, flavor, and overall preferences given by trained panelists [52]. In the study of Desai et al. [23], consumer preferences were not significantly different between traditional strained yogurt and fortified Greek yogurt. However, the results of the quality property and consumer preference test in this study confirmed that the use of *L. gasseri* BNR17 and *L. plantarum* HY7714 as starter cultures could enhance consumers’ preference for Greek yogurt.

## 4. Conclusions

*Streptococcus thermophilus* (ST), *Lactobacillus bulgaricus* (LB), *Lactobacillus gasseri* BNR17, and *Lactobacillus plantarum* HY7714 were used to evaluate the combined effects of various probiotic LAB on the quality and safety aspects of Greek yogurt. The pH and titratable acidity of Greek yogurt made with ST, LB, *L. gasseri* BNR17, and *L. plantarum* HY7714 (T4) was kept relatively constant. In contrast, Greek yogurt made with ST and LB (T1) showed significant changes in pH and titratable acidity (*p* < 0.05), leading to the lowest preference scores for all sensory attributes. Greek yogurt containing *L. plantarum* HY7714 (T3 and T4) had high viscosity, consistent with the results of the viscosity score in the consumer preference test. At 4 °C and 10 °C, the most effective antimicrobial effect against EHEC was observed with T1 due to its low pH and high titratable acidity. At 25 °C, EHEC showed low viability in Greek yogurt containing *L. gasseri* BNR17 and *L. plantarum* HY7714 than in commercial Greek yogurt (*p* < 0.05). Sensory panelists preferred Greek yogurt containing *L. gasseri* BNR17 and *L. plantarum* HY7714 (T4) over other samples. Thus, it is concluded that using probiotic LAB such as *L. gasseri* BNR17 and *L. plantarum* HY7714 as starter cultures for Greek yogurt manufacturing can enhance consumers’ preference and functionality.

## Figures and Tables

**Figure 1 foods-11-03799-f001:**
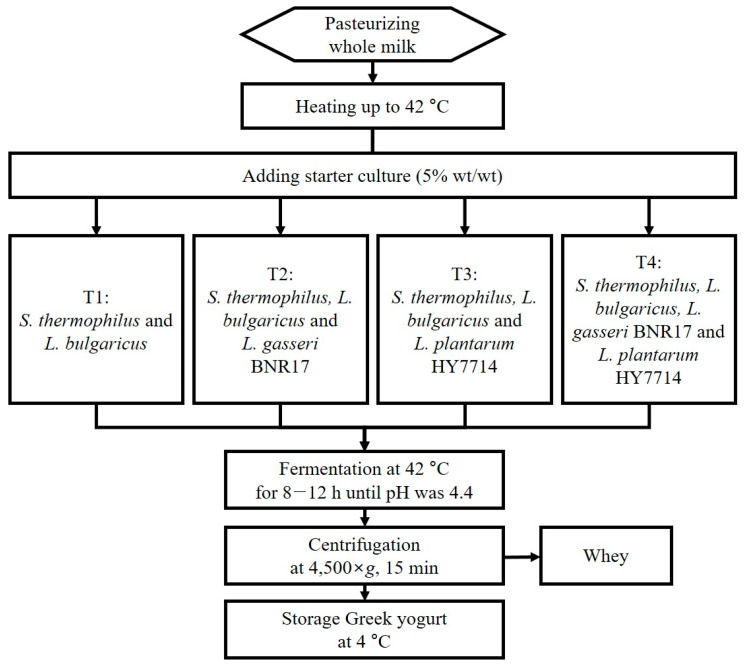
Flow chart of the manufacturing of Greek yogurt with four types of starter culture.

**Figure 2 foods-11-03799-f002:**
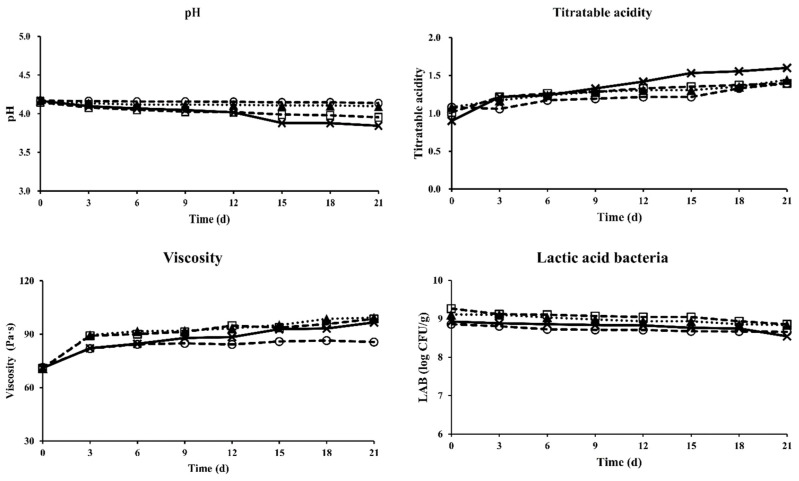
Change of physicochemical characteristics of Greek yogurt with different treatments during storage for 21 days at 4 °C. *S. thermophilus* and *L. bulgaricus* (T1): **x**, *S. thermophilus*, *L. bulgaricus* and *L. gasseri* BNR17 (T2): **○**, *S. thermophilus, L. bulgaricus* and *L. plantarum* HY7714 (T3): ▲, *S. thermophilus, L. bulgaricus, L. gasseri* BNR17, and *L. plantarum* HY7714 (T4): **□**.

**Figure 3 foods-11-03799-f003:**
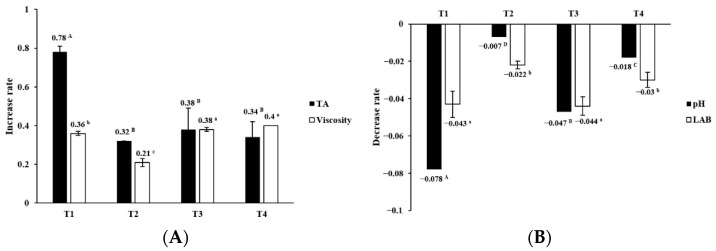
Increase rate of titratable acidity and viscosity (**A**), and decrease rate of pH and LAB (**B**). Decrease rate = (final value − initial value)/initial value, Increase rate = (final value − initial value)/initial value. T1: *S. thermophilus* and *L. bulgaricus*; T2: *S. thermophilus, L. bulgaricus* and *L. gasseri* BNR17; T3: *S. thermophilus, L. bulgaricus* and *L. plantarum* HY7714; T4: *S. thermophilus, L. bulgaricus, L. gasseri* BNR17 and *L. plantarum* HY7714. ^A–D^ Means values in the TA and pH categories with different letters are significantly different (*p* < 0.05) ^a–c^ Means values in the viscosity and LAB categories with different letters are significantly different (*p* < 0.05).

**Figure 4 foods-11-03799-f004:**
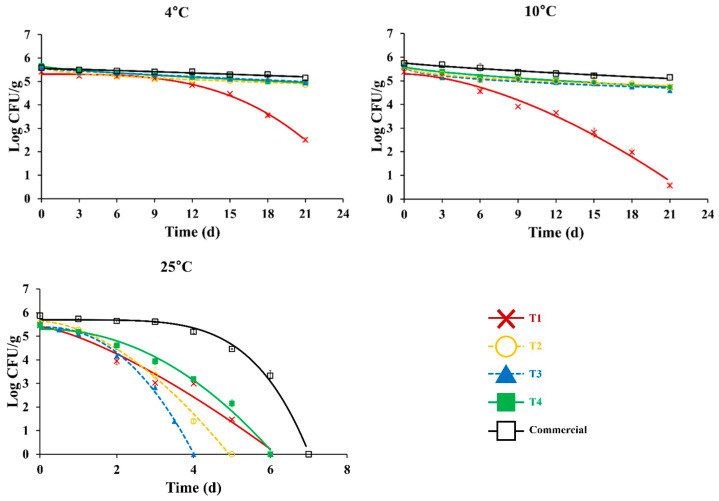
The effect of lactic acid bacteria on the survival of enterohemorrhagic *Escherichia coli* in Greek yogurt at 4, 10, and 25 °C. *S. thermophilus* and *L. bulgaricus* (T1): **X**, *S. thermophilus, L. bulgaricus* and *L. gasseri* BNR17 (T2): **○**, *S. thermophilus, L. bulgaricus* and *L. planatarum* HY7714 (T3): ▲, *S. thermophilus, L. bulgaricus, L. gasseri* BNR17 and *L. planatarum* HY7714 (T4): ■, Commercial Greek yogurt (ILDONG Foodis Plain Greek yogurt): **□**.

**Table 1 foods-11-03799-t001:** Sensory attributes of Greek yogurt.

Attribute	Definition ^1^
Flavor	The tangy and dairy-sour flavor
Sweetness	The basic taste associated with sugar
Sourness	The basic taste associated with acid
Viscosity	The force required to move the spoon back and forth
Creaminess	Smooth texture and behave like a fluid product.
Mouthfeel	The physical sensation created by food in the mouth.

^1^ Terms were adapted from Desai et al. [23], Cayot et al. [24], and Greis et al. [25].

**Table 2 foods-11-03799-t002:** Delta values in Greek yogurt at 4, 10, and 25 °C.

Sample ^1^	Temperature		
	4	10	25
T1 ^1^	15.47 ± 0.11 ^d^	8.30 ± 0.23 ^e^	1.63 ± 0.24 ^d^
T2	40.93 ± 0.91 ^bc^	24.90 ± 1.54 ^c^	1.98 ± 0.06 ^c^
T3	37.30 ± 3.34 ^c^	19.69 ± 2.23 ^d^	1.79 ± 0.05 ^d^
T4	42.04 ± 1.51 ^b^	31.06 ± 5.96 ^b^	2.65 ± 0.18 ^b^
Commercial ^2^	47.76 ± 6.16 ^a^	36.77 ± 2.67 ^a^	5.00 ± 0.07 ^a^

^1^ T1: Fermented by *S. thermophilus* and *L. bulgaricus*, T2: Fermented by *S. thermophilus*, *L. bulgaricus* and *L. gasseri* BNR17, T3: Fermented by *S. thermophilus*, *L. bulgaricus* and *L. plantarum* HY7714, T4: Fermented by *S. thermophilus*, *L. bulgaricus*, *L. gasseri* BNR17, and *L. plantarum* HY7714. ^2^ ILDONG Foodis Plain Greek yogurt. ^a–e^ Means values in the same column with different letters are significantly different (*p* < 0.05).

**Table 3 foods-11-03799-t003:** Sensory scores of Greek yogurts with different starter cultures by Consumer preference test.

Sample ^2^	Sensory Scores ^1^						
	Flavor	Sweetness	Sourness	Viscosity	Creaminess	Mouthfeel	Overall Acceptability
T1	5.03 ± 1.35	4.23 ± 1.47 ^b^	4.37 ± 1.58 ^b^	5.15 ± 1.30	5.18 ± 1.26 ^b^	5.30 ± 1.45 ^b^	4.78 ± 1.56 ^b^
T2	4.98 ± 1.35	4.77 ± 1.21 ^a^	4.75 ± 1.27 ^b^	5.25 ± 1.04	5.40 ± 1.17 ^ab^	5.40 ± 1.18 ^b^	5.20 ± 1.33 ^b^
T3	4.82 ± 1.41	4.77 ± 1.20 ^a^	4.42 ± 1.36 ^b^	5.28 ± 1.01	5.55 ± 1.11 ^ab^	5.57 ± 1.21 ^ab^	5.20 ± 1.27 ^b^
T4	5.08 ± 1.33	5.05 ± 1.47 ^a^	5.30 ± 1.34 ^a^	5.42 ± 1.12	5.68 ± 1.11 ^a^	5.87 ± 0.96 ^a^	5.80 ± 1.13 ^a^

^1^ 1 = dislike very much; 2 = dislike moderately; 3 = dislike slightly; 4 = neither like nor dislike; 5 = like slightly; 6 = like moderately, 7 = like very much. ^2^ T1: fermented by *S. thermophilus* and *L. bulgaricus*, T2: fermented by *S. thermophilus*, *L. bulgaricus* and *L. gasseri* BNR17, T3: fermented by *S. thermophilus*, *L. bulgaricus* and *L. plantarum HY7714*, T4: fermented by *S. thermophilus*, *L. bulgaricus*, *L. gasseri* BNR17 and *L. plantarum* HY7714. ^a,b^ Means values in the same column with different letters are significantly different (*p* < 0.05).

## Data Availability

Data is contained within the article.

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
