# Peer review of "Effect of Probiotic Lactic Acid Bacteria (LAB) on the Quality and Safety of Greek Yogurt"

_foods, 2022, doi:10.3390/foods11233799_

Round 1
Reviewer 1 Report
This study described the effect of some LAB on the quality of Greek yogurt from different aspects. To date, numerous LAB has been isolated and studied, and they must have different effect on the quality of Greek yogurt. My major problem is that why these strains were applied? Except their functional properties, whether they have beneficial effect on the quality of Greek yogurt than the strains that widely used in the commercial products.
Author Response
Lactobacillus gasseri BNR17 and L. plantarum HY7714 have been used as diet pills and healthy skin products, respectively. Both products are expensive. The popularity of Greek yogurt is increasing due to its health effect. We would like to know whether these strains can be used to make Greek yogurt. We are mostly concerned about whether consumers like this product or not as yogurt. This work focuses on the possibility of using both strains as a starter culture for Greek yogurt manufacturing. The results of the sensory evaluation show that Greek yogurt containing L. gasseri BNR17 and L. plantarum HY7714 (T4) received the highest preference score among samples, including Greek yogurt made with strains (S. thermophilus and L. bulgaricus) that are widely used in commercial products.

Reviewer 2 Report
After analyzing the manuscript, I have some remarks and comments:
1. Paragraph 60-69 separate and insert another subsection. Entire section 2.1. modified to indicate the preparation of yogurt with probiotics and the effect of LAB probiotics on Enterohaemorrhagic E. coli. The authors, as they indicate in the title, examine "safety". These things have to be separated.
2. In Figure 1 some information is missing: how much these bacteria were added and to what pH value the fermentation was carried out.. Please add it.
3. In Figure 3, markings A and B are missing in the text.
4. What was the fat content and total solids in manufacturing of Greek Yogurt? Please indicate the composition of manufacturing Greek yogurt.
5. In section 2.3. the description of the electrode used is missing. Please complete.
6. The entire subchapter 2.4. /Sensory evaluation/ should be heavily modified. Based on which method was the sensor evaluation carried out? Please also explain what you compared to “the flavor, sweet taste, sour taste, viscosity, creamy, mouthfeel test”. What was the model for you? The flavor itself is also a general statement. What do you mean by that term? Likewise, when assessing the "mouthfeel taste" parameter, what did it refer to? There is no appropriate table with attribute type and attributes description. Please complete it!!.
7. What bacteria were in commercial Greek yogurt? When we take something to compare, we describe the bacteria and not the name of the commercial product. Please add an appropriate sentence (after the sentence in Line 128).
After analyzing the entire manuscript, some very important analyzes are missing. Especially:
A. Detailed analysis of the texture of yogurt. In particular, the parameters: firmness, consistency, cohesiveness. Can the authors give what these parameters looked like? Viscosity alone is, in my opinion, insufficient.
B. Analysis of the color of the yogurts obtained and comparing them with commercial yogurt. Especially the designation: chrome (C *), white index (WI), and yellowing index (YI). Are the authors able to provide information on this?
When assessing "quality" all aspects should be taken into account, including instrumental evaluation. Color and texture are important quality characteristics. The general characteristics that the authors made in paragraphs 282-301 are not sufficient in my opinion. Please complete these analyzes for your yogurts.
C. Have the authors forgotten about affiliation, abstract and references?
Reviewer 3 Report
The Authors investigated the effects of probiotic LAB on quality, sensory, and microbiological characteristics of Greek yogurt. Greek yogurt fermented with S. thermophilus, L. bulgaricus, L. gasseri BNR17, and L. plantarum HY7714 showed superior physicochemical properties and received the highest preference score from sensory evaluation among samples. Interesting and valuable paper. It has application possibilities. The addition of L. gasseri BNR17 19 and L. plantarum HY7714 as starter cultures can enhance the microbial safety and sensory properties of Greek yogurt. The topic was investigated by previous researchers, but different combination of LAB were used. Other researchers were focused mainly on quality of yoghurt or microbiological safety. In presented paper both important issued were addressed in the same paper. Presented conclusions are consistent with presented results. The references are appropriate. Well written, English level is acceptable.
9 should be: Greek yogurt is known as a healthy snack that can increase lean muscle mass and decrease body fat.
105-106 viscosity should be in mPas and not Pas
Figure 2....viscosity should be in mPas and not Pas
Round 2
Reviewer 1 Report
I think the aurtors did los of sufficient work about the effect of specific lactobaccillus strain on the greek yogurt, and some neccessary index have been measure. But to me, every bacterial strain is specific, and they can bring specific effect on the milk products, including flavor, microviscroty and so on. From the novelty of article, I dont think it bring a lot of suprising to me.
Reviewer 2 Report
There is still no description of the terms used (in 2.4. Sensory evaluation) - what was the reference? What made someone give 1 = very unpleasant and 7 = very pleasant?
